

# Computer simulation of Cerebral Arteriovenous Malformation—Validation analysis of hemodynamics parameters

Y. Kiran Kumar[1,2], Shashi Bhushan Mehta[2] and Manjunath Ramachandra[2]

[1] Philips Research, Philips Innovation Campus, Bangalore, India
[2] Manipal University, Manipal, India

## ABSTRACT

**Problem**. The purpose of this work is to provide some validation methods for evaluating the hemodynamic assessment of Cerebral Arteriovenous Malformation (CAVM). This article emphasizes the importance of validating noninvasive measurements for CAVM patients, which are designed using lumped models for complex vessel structure.

**Methods**. The validation of the hemodynamics assessment is based on invasive clinical measurements and cross-validation techniques with the Philips proprietary validated software's Qflow and 2D Perfursion.

**Results**. The modeling results are validated for 30 CAVM patients for 150 vessel locations. Mean flow, diameter, and pressure were compared between modeling results and with clinical/cross validation measurements, using an independent two-tailed Student $t$ test. Exponential regression analysis was used to assess the relationship between blood flow, vessel diameter, and pressure between them. Univariate analysis is used to assess the relationship between vessel diameter, vessel cross-sectional area, AVM volume, AVM pressure, and AVM flow results were performed with linear or exponential regression.

**Discussion**. Modeling results were compared with clinical measurements from vessel locations of cerebral regions. Also, the model is cross validated with Philips proprietary validated software's Qflow and 2D Perfursion. Our results shows that modeling results and clinical results are nearly matching with a small deviation.

**Conclusion**. In this article, we have validated our modeling results with clinical measurements. The new approach for cross-validation is proposed by demonstrating the accuracy of our results with a validated product in a clinical environment.

Corresponding author
Y. Kiran Kumar,
kiran.kumary@philips.com

## INTRODUCTION

Cerebral Arteriovenous Malformation (CAVM) is a neurovascular malformation. The cerebral vasculature of a healthy individual consists of arteries and veins which are connected by capillaries. In CAVM the capillaries are absent, resulting in a tangled cluster of vessels. The vessel geometry in CAVM is complex in nature. The CAVM patient is affected by hemodynamics changes. Invasive techniques are the current clinical procedure to measure the hemodynamics of CAVM. The invasive techniques are risky to patients,

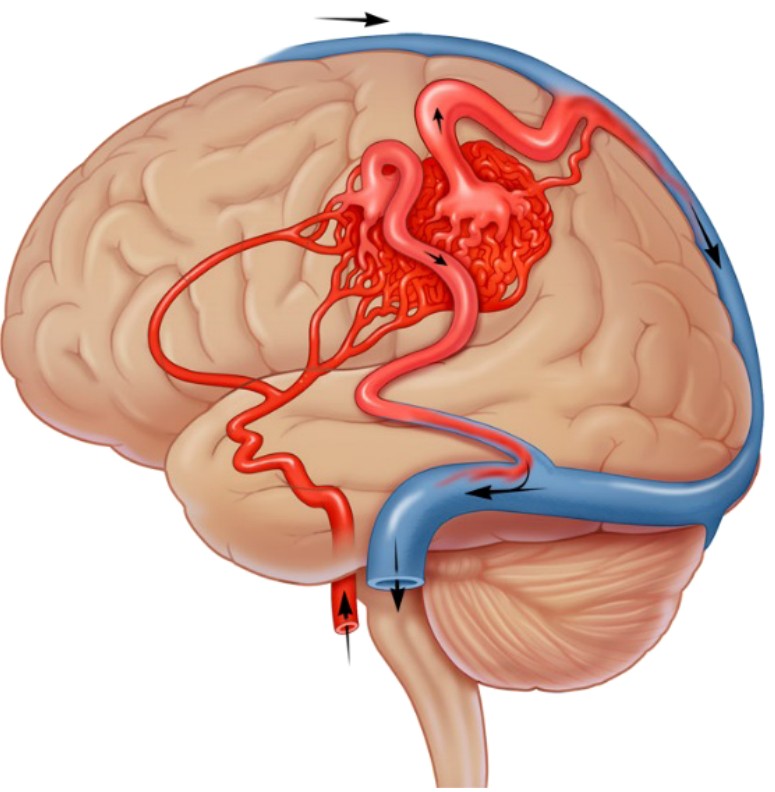

**Figure 1** Cerebral Arteriovenous Malformation (CAVM).

since CAVM can rupture. Figure 1 shows the complex structure of CAVM. The gold standard imaging for CAVM is the Digital Subtraction Angiogram (DSA); Fig. 2 shows the CAVM—DSA image (*Liu, 1993*; *Saleh, 2008*; *Yasargil, 1987*).

In this article, we have demonstrated the accuracy of our modeling results with clinical measurements and with cross-validation techniques. We replicate actual patient conditions by simulating similar patient conditions using Matlab simulation. Lumped models are created and simulated using different signal combinations, which help to validate our results with clinical measurements.

## METHODOLOGY

The non-invasive technique to measure hemodynamics in the complex vessel in CAVM is based on a lumped model. In this article, we focus on the different validation techniques to demonstrate the accuracy of our modeling results. The non-invasive measurements are validated in two ways: Invasive technique and Cross validation. The complex vessel structures are formed by combinations such as bifurcation, vessel feedback, vessel deformation, vessel collapsing, vessel bending, tortuosity, and so on. The analysis for the complex vessel structure is performed using lumped modeling (*Kumar, Mehta & Ramachandra, 2014*; *Kumar, Mehta & Ramachandra, 2013a*; *Kumar, Mehta & Ramachandra, 2013b*). The output pressure measurement of the lumped model is validated

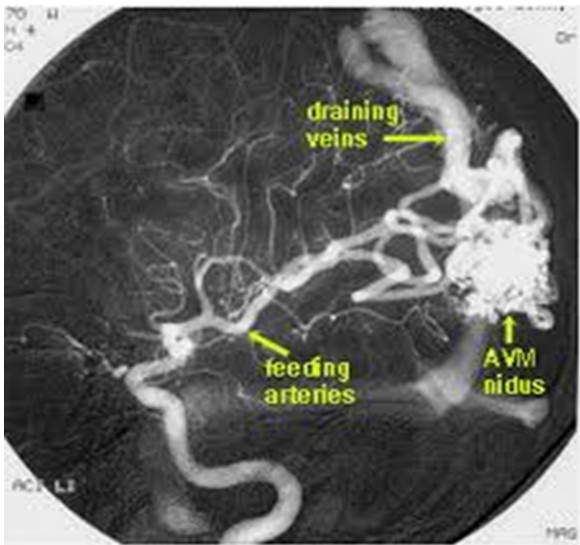

**Figure 2** Digital subtraction angiogram of CAVM image.

with invasive and cross-validation techniques for 30 CAVM patients, with 150 vessel locations.

## Invasive techniques

The clinical procedure to acquire data from a patient is to insert the catheter from the femoral artery by performing a single and multi-puncture in the femoral artery. The catheter is 0.08 inch /0.2 mm in width, with a 200 mm length (*Saleh, 2008*). The catheter wire was propagated slowly following vascular structures of different diameters and bends till it reached the respective CAVM (*Valavanis, Pangalu & Tanaka, 2005*). The catheter is navigated slowly through the path until it reaches the CAVM vessels (*Valavanis, Pangalu & Tanaka, 2005*). The KMC Manipal has approved the ethical clearance for this study.

The pressure bag has pressure sensors that are externally connected to the guided catheter. The pressure bag readings are shown in the patient monitor system. After reaching the required vessel location, the clinician measures the pressure value from the patient monitor. The patient monitor also shows ECG, heart rate, respiratory rate along with pressure value. Figure 3 shows a patient monitor along with pressure values obtained from the Cath Lab in KMC Manipal. The pressure is measured for various arteries—left external cerebral artery, internal carotid artery, posterior cerebral artery, middle cerebral artery, near Nidus (*Standring, 2008*). This procedure is used to measure pressure at various vessel locations in the Cath Lab. The pressure values obtained by the clinical procedure are taken as references for the validation of the modeling results.

## Cross validation

The cross validation technique is a type of validation where the modeling results are cross validated with equivalent software which produces same results. In our article, we validated our results with Philips proprietary software such as Q-Flow and the 2D Perfusion analysis software.

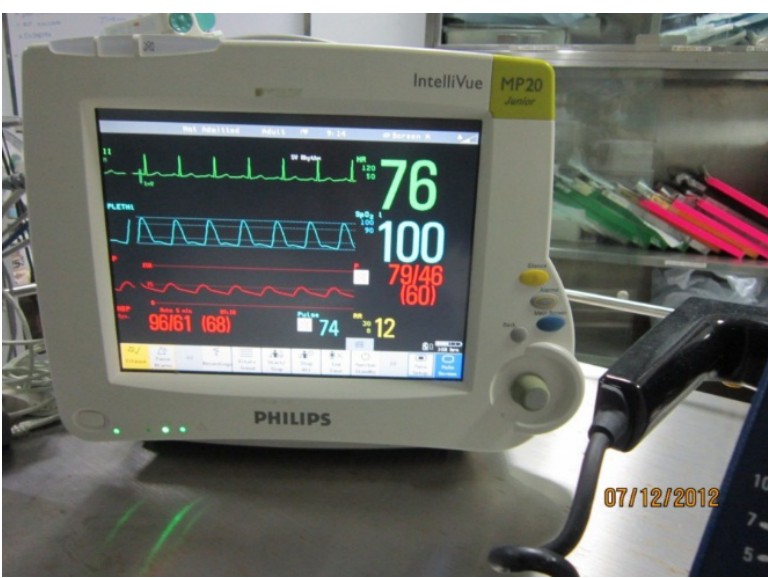

**Figure 3** **Clinical pressure measurements.** Courtesy: Kasturba Medical College & Hospital, Manipal.

### Q-Flow software

The validations of complex geometries, feeding arteries are performed using a Philips proprietary product called Qflow. Qflow is developed and validated by Philips Healthcare. Qflow is commonly used in hospitals for clinical diagnosis and treatment, and is a validated software accepted by clinicians (*Lotz, 2002*; *Kondo, 1991*). The Qflow application requires MR Angiogram (MRA) data (Fast Field Echo (FFE) & Phase) for processing. MRA data of CAVM patient with different phase information was obtained from the KMC hospital. The MR Angiogram is an imaging technique used to obtain phase and flow analysis of the patient. The MRA for cerebral patients gives cerebral flow parameters. Using the Qflow software, we obtain the velocity of the blood flow. The velocity is converted to pressure, which is used for our validation analysis. Our study has approximated conversion between the velocities to pressure. This approximation results in loss of accuracy, which is analyzed in the Results section.

### 2D Perfusion software

The modeling results are validated with the Philips proprietary Cath Lab software known as 2D Perfusion. The input data is by DSA image. The Philips validated 2D Perfusion software is a software product that provides functional information about tissue perfusion based on a digital subtraction angiography (DSA). It can visualize multiple parameters related to perfusion.

*Statistical analysis.* Mean flow, diameter, and pressure were compared between modeling results and with clinical/cross validation measurements, using an independent 2-tailed Student $t$ test; refer to Appendix S1 for $t$ test results. Exponential regression analysis was used to assess the relationship between blood flow, vessel diameter, and the pressure between them. Univariate analysis was used to assess the relationship between the vessel

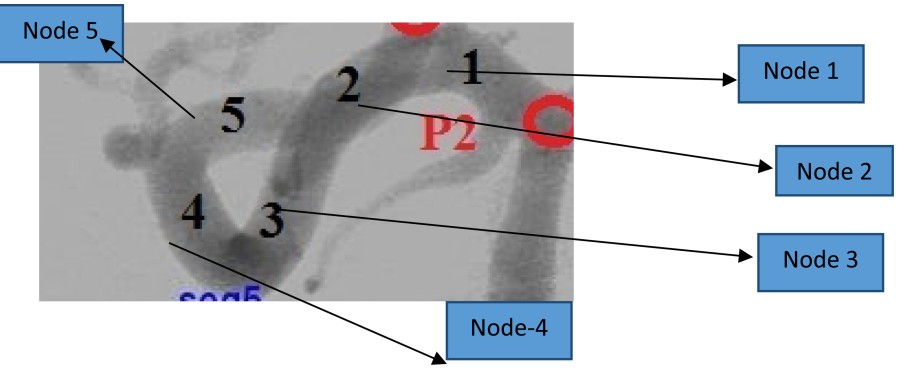

**Figure 4 Complex vessel structure.**

diameter, the vessel cross-sectional area, the AVM volume, and the AVM pressure; the AVM flow results were performed with linear or exponential regression. Two-way tables were verified using Fisher's exact test, and regular logistic regression was used to evaluate the association between pressure and diameter variation in the vessel. All analyses were performed with SPSS (Version 22; IBM Inc.) (*Anna et al., 2014*).

Node voltage outputs were expressed as mean value ± standard deviation. A total of 30 AVM patients were studied with evaluation of 150 vessels locations as node point. The statistical analysis for various node output of loop structure is shown in Table 1. The statistical analysis shows that the average error rate is less than 0.05 and the mean square error is less when compared to other simulation results. A *P*-value <0.05 was considered significant (refer Appendix S1 for results). The standard deviation for each node is less than 0.05, compared to other simulation results. This shows that the proposed model based on a non-invasive technique has advantages over other simulation techniques.

## RESULTS AND DISCUSSION

### Invasive validation

The invasive hemodynamics measurement inside NIDUS is risky, due to the complex geometric structure of NIDUS. However, with help from clinicians in the Cath Lab at KMC Manipal, we were able to measure pressure values near locations of Nidus using a guided micro catheter. The measured locations are the external carotid artery, the internal carotid artery, and the posterior cerebral artery. The simulation is performed for the complete path of node1 to node5 (refer to Fig. 4). The pressure measurements for the loop structure are shown in Table 2. The model is simulated with different signal magnitude variations.

Table 2 shows modeling results are validated against clinical measurements. The input voltage/pressure used for simulation is 80 mmHg/0.8 volts. Each node represents a corresponding cerebral vessel location. Each node is modeled using lumped elements and the corresponding node outputs are compared against clinical measurements. The percentage deviation in Table 2 represents the amount of percentage difference between the modeling results and clinical results.

**Table 1  Statistical analysis for various node outputs of the loop structure.**

| Quantification parameters | Node1 output voltage | Node2 output voltage | Node3 output voltage | Node4 output voltage | Node5 output voltage |
|---|---|---|---|---|---|
| Count | 12 | 12 | 12 | 12 | 12 |
| Minimum | 3.4159 | 3.4159 | 3.4159 | 3.4159 | 3.4159 |
| Maximum | 4.3 | 4.3 | 4.3 | 4.3 | 4.3 |
| Sum | 19.42477 | 19.42477 | 19.42477 | 19.42477 | 19.42477 |
| Mean | 4.4159 | 3.4159 | 3.4159 | 3.4159 | 3.4159 |
| Median | 3.14159 | 3.14159 | 3.1414 | 3.15 | 3.15 |
| Mode | N/A | N/A | 3.1414 | N/A | N/A |
| Range | 0 | 0 | 0.00018 | 0 | 0 |
| Interquartile range | 0 | 0 | 0.00018 | 0 | 0 |
| Standard deviation (range) | $5.43896E-16$ | $5.43896E-16$ | $5.43896E-16$ | 0 | 0 |
| Standard deviation (Population) | $5.44089E-16$ | $5.44089E-16$ | $5.44089E-16$ | 0 | 0 |
| Variance (Sample) | $3.95823E-31$ | $3.95823E-31$ | $3.95823E-31$ | 0 | 0 |
| Variance (Population) | $1.97215E-31$ | $1.97215E-31$ | $1.97215E-31$ | 0 | 0 |
| Sum of squares | 39.60876318 | 39.60876318 | 39.60876318 | 39.60876318 | 31.7675 |
| Mean squared error | 9.869587728 | 9.869587728 | 9.868770939 | 9.9225 | 9.93 |
| Root mean squared error | 3.14159 | 3.14159 | 3.141460001 | 3.15 | 3.151 |
| Mean absolute deviation | $4.44089E-16$ | $4.44089E-16$ | $8E-05$ | 0 | 0 |
| Skewness | 2.449489743 | 2.449489743 | 1.732050808 | 65,535 | 65,535 |
| Standard error of skewness | 1.224744871 | 1.224744871 | 1.224744871 | 1.224744871 | 1.224744871 |
| Excess kurtosis | 65,535 | 65,535 | 65,535 | 65,535 | 65,535 |
| Standard error of kurtosis | 65,535 | 65,535 | 65,535 | 65,535 | 65,535 |
| Jacque–Bera test stat | 65,535 | 65,535 | 65,535 | 65,535 | 65,535 |
| Durban–Watson test stat | 0 | 0 | $1.6416E-09$ | 0 | 0 |

**Table 2  Loop structure pressure measurements and analysis.**

| Nodes | Input voltage pressure = 0.8 volt / 80 mmHg | | |
|---|---|---|---|
| | Measured value | Clinical results | Percentage deviation % |
| Node1 | 0.72v/72 mmHg | 0.74v/74 mmHg | 2.7 |
| Node2 | 0.7v/70 mmHg | 0.72v/72 mmHg | 2.7 |
| Node3 | 0.57v/57 mmHg | 0.60v/60 mmHg | 5 |
| Node4 | 0.52v/52 mmHg | 0.55v/55 mmHg | 5.4 |
| Node5 | 0.47v/47 mmHg | 0.50v/50 mmHg | 6 |

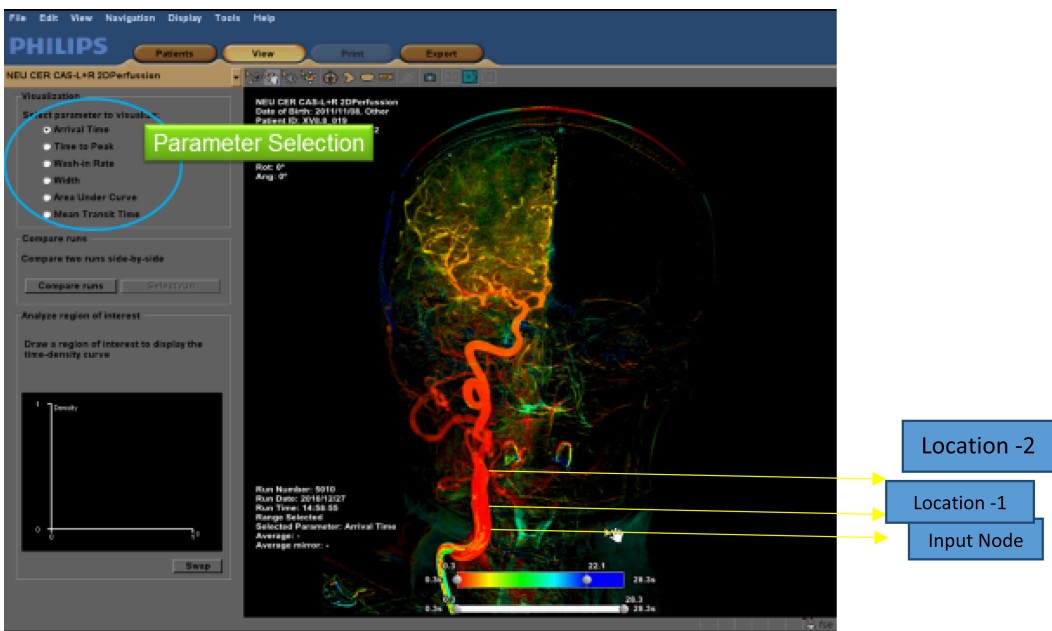

**Figure 5** Qflow analysis with node locations.

**Table 3** Qflow validation with modeling results.

| Vessel location as per Fig. 5 | Flow outputs peak velocity as per Qflow outputs in volts | Electrical network— modeling output | Deviation % |
| --- | --- | --- | --- |
| Input | 0.02 volts—input voltage (Qflow initial velocity – max 200 cm/s) | 0.0187 volts | 6.5 |
| Location 1 | 0.012 volts | 0.01 volts | 0.001 |
| Location 2 | 0.003 volts | 0.0225 | 0.0195 |

## Cross validation techniques
### Q-Flow validation

The study is validated by comparing against Qflow results with modeling results. The Qflow processing results are velocity components for specific node/regions. The lumped model is created for a specific node/region and is simulated for same input used in Qflow application. Figure 5 shows an MRA image of a CAVM patient with velocity results for the drawn region of interest in the cerebral vascular region. Table 3 compares modeling results against Qflow results along with amount of difference between them. The Qflow validation analysis is performed for each phase acquisition of MRA of CAVM patient. The table shows various locations such as location 1, location 2, depicts the pressure measurements at corresponding location for various phases.

Pre-requisite: Conversion of maximum velocity to volts

Table 4 shows pressure values obtained from each phase of MRA flow study compared against with our modeling results. The percentage deviation shows the amount of variation

**Table 4  CAVM-MRA flow study for various phases & CSF region.**

| Vessel locations as per Fig. 5 | Flow outputs mean velocity (in volts) | Electrical network modeling output | Deviation % |
|---|---|---|---|
| **Phase 3:** | | | |
| Input | 0.03 | 0.0278 | 0.0022 |
| Location 1 | 0.015 | 0.01 | 0.005 |
| Location 2 | 0.01 | $2 \times 10^{-4}$ | 0.0098 |

| Vessel locations as per Fig. 5 | Flow outputs mean velocity (in volts) | Electrical network modeling output | Deviation % |
|---|---|---|---|
| **Phase - 8** | | | |
| Input | 0.35 | 0.337 | 0.013 |
| Location 1 | 0.28 | 0.268 | 0.012 |
| Location 2 | 0.19 | 0.178 | 0.012 |

between modeling results with the Qflow pressure results. The reason for the deviation is due to the conversion factor from velocity to pressure values.

Conversion factor:

Input location: mean $-$ 0.4 cm/s $=$ 0.03 volts $-$ input voltage

Location 1- 0.2 cm/s–0.015 volts

Location 2- 0.1 cm/s–0.01 volts.

### 2D Perfusion validation

2D Perfusion can be used for the identification of perfusion alterations in blood vessel perfusion behavior, e.g., in CAVM. The following are the list of parameters that are used for validation with modeling outputs:

- Model fit to the time density curve:

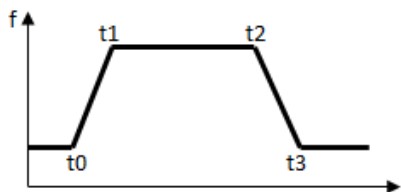

- Time of Arrival $= t0$
- Time to Peak $= t1 + t2$
- Wash-in rate: $\int_{t1}^{t2} (t2 - t1)f(t0) - \int_{t1}^{t2} f(t)\,dt)/(t2 - t1)(tp - t0)$
- Width $= \frac{t2+t3}{2} - (t0 + t1)/2$
- Area under Curve:

$$A = \int\int_{to}^{t3} k(f(t) - f(0)dt$$

- Mean Transit Time:

MTT $= \sum_{i=0}^{3} f(ti)/\sum_{i=0}^{3} f(ti).$

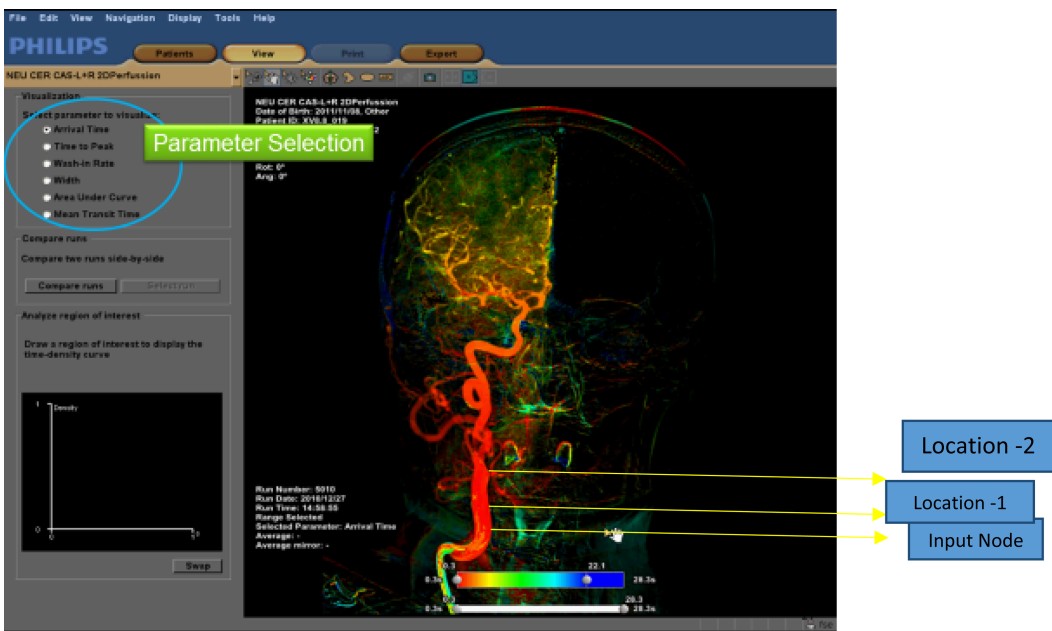

**Figure 6** 2D Perfusion analysis.

**Table 5** 2D perfusion–cross validation.

| Vessel location as per Fig. 6 | Cerebral blood volume (pressure in volts) | Electrical network modeling output | Deviation % |
|---|---|---|---|
| Input | 0.12 volts | 0.115 volts | 4.1 |
| Location 1 | 0.22 volts | 0.209 volts | 5 |
| Location 2 | 0.43 volts | 0.415 volts | 3.4 |

These clinical parameters are the output of perfusion software. These parameters are converted in to the electrical equivalent for validation analysis, the details are as follows:

- Cerebral Blood flow (CBF) ∼ Wash in Rate- Flow rate ∼ current
- Cerebral Blood Volume (CBV) ∼ Area under Curve/Width–velocity ∼ pressure
- Mean Transit Time (MTT) ∼ CBV/CBF = (Area under Curve/Width)/Wash in Rate- Friction coefficient ∼ Resistance.

The model is validated with DSA data from 15 CAVM patients. The results are closely matched, and have an accuracy of 85%. The accuracy of modeling results is affected due to approximations in conversion of clinical to electrical parameters Fig. 6 shows a snapshot of 2D Perfusion along with clinical parameters

Table 5 represents cross validation analysis for various vessel locations such as location-1, location-2 (refer Fig. 6). Modeling results are compared against 2D Perfusion software. The amount to percentage deviation is calculated for each cross validation comparison.

## DISCUSSION

The clinical procedure to measure hemodynamics in CAVM is an invasive procedure. The current procedure is risky, as the catheter may rupture and can cause patient death (*Erzhen & William, 1998*; *Wayne & Brahm, 2008*). The researchers explained different models based on invasive techniques for hemodynamics analysis (*Cattivelli et al., 2008*) but were limited by the measurement of radius calculations for specific arteries. *Kuebler et al. (1998)* analyzed the regional cerebral blood flow based on the non-invasive technique, however limited by cerebral circulation. *Kienzler et al. (2015)* analyzed various methods for validation of noninvasive pressure measurements, but limited by data points.

The proposed non-invasive methodology addresses the clinical procedure to measure hemodynamics by simulating the actual patient conditions using a lumped model. The modeling results are validated with the clinical invasive measurements. Our results show that simulated results are an approximate match with the actual clinical measurements. The reason for the deviation is due to the conversion factor from velocity to pressure values. A total of 30 CAVM patients and 150 vessel locations were validated with the invasive measurements and with cross validation (Qflow). The statistical analysis shows that the mean square error rate for 150 vessel locations is less than 0.05, shows a statistically significant evidence. The modeling results is approximately matching with Qflow results. The reason for the deviation is due to conversion factor from pressure to velocity parameter.

In the validation of 2D Perfusion, the data is limited to 15 DSA images, as this software requires a specific type of DSA acquisition to process. With the 15 patients, we validated and quantified our modeling results with 2D Perfusion. The results are nearly matching, with an accuracy of 85%. The reason for deviation in the modeling results with 2D perfusion is due to the approximation of parameters used for the modeling.

Cross validation is a novel approach for CAVM validation. Qflow and 2D Perfusion software's implementation is based on mechanical simulation. The lumped modeling results, which nearly match with those of the Philips proprietary software, confirm the matching of results between electrical and mechanical simulation. To reach evidence >98%, we require more data for our validation work. The limitation of this work is the use of multiple approximated conversions of specific applications compared against modeling results. This work can be extended for different geometry using three dimensional volume data and require optimization of conversion factors to increase the accuracy of modeling results.

## CONCLUSION

In this article, we have validated CAVM modeling results created using lumped networks with clinical measurements and with cross-validation techniques. A new approach for cross-validation is proposed in this article. The modeling results demonstrated the accuracy of the method with a validated product in a clinical environment. The results were validated for 30 CAVM patients with 150 vessel locations, and this validation shows nearly matching results compared to the invasive measurements and with the Philips proprietary validated software. This method seems to be safe and reliable.

### Funding

The authors received no funding for this work.

### Competing Interests

The authors declare there are no competing interests.

### Author Contributions

- Y. Kiran Kumar conceived and designed the experiments, performed the experiments, analyzed the data, contributed reagents/materials/analysis tools, wrote the paper, prepared figures and/or tables.
- Shashi Bhushan Mehta reviewed drafts of the paper, reviewer of paper.
- Manjunath Ramachandra reviewed drafts of the paper.

### Ethics

The following information was supplied relating to ethical approvals (i.e., approving body and any reference numbers):

Ethics board of Manipal University, Manipal, India.

### Data Availability

The raw data has been supplied as a Supplemental File.

### Supplemental Information

Supplemental information for this article can be found online at http://dx.doi.org/10.7717/peerj.2724#supplemental-information.

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
