# Peer review of "Computer simulation of Cerebral Arteriovenous Malformation—validation analysis of hemodynamics parameters"

_PeerJ, doi:10.7717/peerj.2724_

## Round 0.1 · original submission · Major Revisions

· Academic Editor

Major Revisions

Dear Authors,

Please note that a 'major revision' decision by both peer reviewers has been given to the submitted manuscript which will require detailed changes into the format and statistics of the study as well.

·

Basic reporting

1. The authors do not follow Standard Sections in the manuscript. Neither do they use headings in structured abstract.
2. The wording in the whole text including abstract requires extensive revisions.
3. The quality of Figure 3 is poor.
4. Table 2 is missing, but Figure 5 duplicates.
5. There is no Discussion. The authors can discuss the differences among various invasive and noninvasive techniques, giving evidence from literatures. The reason why non-invasive methods can confidently replace the invasive one may also be interesting.

Experimental design

1. The Cross Validation methods, including Q-flow software and 2D-Perfusion Software, require more detailed elaboration, not just their origins. For instance, the authors mention that Qflow application requires MR Angiogram (MRA) data for processing (“what”), but the role of MRA data (“why” and “how”) in achieving Qflow application is not explained further.

2. The authors should have to explain the modeling results more clearly.
3. There are posterior cerebral artery and middle cerebral artery, instead of posterior and middle “carotid” arteries.

Validity of the findings

1. The “Location 1” and “Location 2” in Tables 3, 4 and 5 are never mentioned in the text, legends or figures, leading to uncertainty of their locations.
2. In the second Figure 5, do “MPA” and “LPA” really mean “Main pulmonary artery” and “Left pulmonary artery” in cerebral vascular region?
3. I can’t see where the sensitivity of 95% and specificity of 96% are derived from. The authors seem to jump to the conclusion abruptly.

Additional comments

To my understanding, the authors try to replace invasive assessment of vascular characteristics of cerebral arteriovenous malformation with noninvasive methods. This is what many neuroscientists in the world are making efforts to do. If they intend to claim that their model works, the followings may be considered:
1. fluent English wording and accurate arrangement of tables and figures;
2. explicit data presentation with pertinent positive and negative results;
3. more solid foundation of literature review;
4. elaboration of the advantages and disadvantages of their model and the methods they use for validation;
5. concise discussion of the clinical applications and their future work.

Reviewer 2 ·

Basic reporting

The authors provide some validation methods to evaluate the hemodynamic assessment of Cerebral Arteriovenous Malformation (CAVM) based on clinical measurements and cross-validation techniques. The results were validated for 150 vessel locations with 30 CAVM datasets.

In general, the article is hard to follow as the writing is not clear.

There are also many grammatical as well as sentence structure errors. For example, line 16 – “…results are validated for 150 vessel locations showed significantly results compared to…” – what does significantly results mean?

There are 3 authors, numbered 1, 2 and 3 but only 1 and 2 are defined. Which department does the third author belong to?

The caption for tables is supposed to appear above the tables, not below.
References are not according to PeerJ format as stated under “Instructions for Authors”.

Experimental design

The output pressure measurement of the model is validated with invasive and cross-validation techniques (line 46/47). However, the authors did not state how many patients were involved.

150 vessel locations of how many patients?

30 CAVM datasets from how many patients?

Any ethical approval for this study?

Line 189 – The 2D perfusion cross validation - the model was validated with 15 DSA of CAVM patients. Why not 30 CAVM patients?

Validity of the findings

Line 229 – A total of 2 AVM patients were studied with evaluation of 150 vessels location. This was not mentioned in the introduction or methodology, but only towards the end, under the Results section.

Explain further on Table 1 results.

Where is Table 2?

Tables 3 and Table 4 with phases 3, 5, 8 and 11 show the parameters taken. What are the results?

The conclusion is not clear.

Additional comments

Overall presentation of this study is not well-structured, with missing and confusing tables.

Neither is it well-rendered.

---

## Round 0.2 · Major Revisions

· Academic Editor

Major Revisions

Dear Authors,

The two peer reviewers have raised numerous issues that need to be looked at seriously before it will be accepted by them for publication. I hope all comments will be looked into to make this manuscript suitable to be published in PeerJ.

·

Basic reporting

1. The authors still do not follow Standard Sections in the manuscript. The Discussion section should be separated from Results. Neither do they use headings in structured abstract.
2. The wording in the whole text including abstract still requires extensive revisions. The contents are mostly the same as first submission.
3. The 3rd author comes from the same institution as 2nd one, so he/she should be marked as 2, indicating Manipal University Manipal.
4. In the in-text references, it seems that only the last names, not the full names, of authors need to be listed. In Line 60-61, for four or more authors, abbreviate with ‘first author’ et al. (e.g. Ondra et al., 1990).
5. The Discussion Section should cover more the differences among various invasive and noninvasive techniques, giving evidence from literatures. The details of various approaches can be elucidated.

Experimental design

1. The Cross Validation methods, including Q-flow software and 2D-Perfusion Software, require more detailed elaboration. For instance, in Line 92-93, the authors mention that “Using Qflow software, velocity of the blood flow is obtained and converted to pressure, which is used for our validation analysis. 
” How is the velocity converted to pressure? Is there any loss of accuracy during the conversion?
2. The authors should explained the modeling results more clearly.
3. There are posterior cerebral artery and middle cerebral artery, instead of posterior and middle “carotid” arteries. There are no External and Internal Cerebral arteries.

Validity of the findings

1. In Line 122-123, “The deviation is within acceptable range as per clinicians after validating with visual inspection”. What is acceptable range? Is “visual inspection” a convincing standard to examine the deviation? Also in Line 151, “The deviation is within acceptable range of ±8%”. How to define ±8% as acceptable range?
2. Table 5 lists only the raw statistical data, but no further analysis or discussion is available. How do the authors make the conclusion out of the raw data?

Additional comments

In the revised manuscript, the authors keep trying to replace invasive assessment of vascular characteristics of cerebral arteriovenous malformation with noninvasive methods. However, the writing is still hard to understand and the authors’ ideas are difficult to reveal.

Previous suggestions are not well followed:
1. fluent English wording and accurate arrangement of tables and figures;
2. explicit data presentation with pertinent positive and negative results;
3. more solid foundation of literature review;
4. elaboration of the advantages and disadvantages of their model and the methods they use for validation;
5. concise discussion of the clinical applications and their future work.

Please think from a layman's viewpoint and do not jump from raw data to conclusion too quickly. The readers could not catch the point easily, esp. with a poorly-structured writing.

Reviewer 2 ·

Basic reporting

No comments

Experimental design

No comments

Validity of the findings

Table 5 –Should have some explanations but there is none even though it has more than 20 rows. Use the text component to guide readers on the results.

The discussion is too brief.

Additional comments

Although the contents have been revised accordingly and there are slight improvements, there are still grammatical errors throughout the manuscript.

---

## Round 0.3 · Major Revisions

· Academic Editor

Major Revisions

Dear Authors,

Please heed the suggestions from the Peer Reviewer to revise your manuscript and sent the manuscript for Professional English editing (if you choose to do so then please send us proof of Professional English Editing when you resubmit your manuscript in the next round)

Reviewer 2 ·

Basic reporting

No comments

Experimental design

No comments

Validity of the findings

Line 235 – Normally statistical analysis is mentioned in the methodology section. Line 238 to 247 should be in the Methodology, not Results section.

Where is the result for the independent t test?

Line 255 mentioned a p-value < 0.05 considered as significant, but where is the result?

Additional comments

There are slight improvements in the manuscript. However, the abstract is still not clear, and author has not followed the format stated by peerJ for in-text citations and references.

Abstract - no results were mentioned in the results section. Author stated the methods, not displaying results. Discussion section – repetition of methods and result, not discussion.

Please refer to Instructions for Authors for in-text citations:-

For three or fewer authors, list all author names (e.g. Smith, Jones & Johnson, 2004). For four or more authors, abbreviate with ‘first author’ et al. (e.g. Smith et al., 2005).

Line 40 – [Liu 1993; Omar Saleh 2008; Yasargil MG 1987] should be cited as (Liu, 1993; Saleh, 2008; Yasargil, 1987).

Line 57, 58, 59 – wrong format of in-text citation. References by the same author in the same year should be differentiated by letters (Smith, 2001a; Smith, 2001b).

Line 267 - in-text citation.

References - please follow Instructions for Authors:-

Overall from line 302 – 344, references are not sorted accordingly. The References Section should be sorted by Author, Year, Title.

Line 302 – 344, references are not listed according to peerJ standard.

For example, line 304 should be listed as –
Salleh O. 2008. Arteriovenous Malformation, complications, and perioperative anesthetic management. M.E.J. Anesth 19 (4): 737-56.
For example, line 308 should be listed as –
Kumar YK, Mehta S, Ramachandra M. 2014. Cerebral Arteriovenous Malformation Modelling. Advanced Science, Engineering and Medicine. 6(1): 105-107.

Reference- line 316 repeats line 304

Reference - line 318 repeats line 305

Line 318 and 319 – 2 references in one.

Line 304 and 305 – 2 references in one.

Line 338, 340 and 342 – list all authors here, not et.al. Each journal reference should be listed using this format: the full list of Authors with initials. Publication year. Full title of the article. Full title of the Journal, volume: page extents. DOI (where you have it)

Line 69 – Yasargil et. al, 1990 is not listed in the reference list.

There are still grammatical errors such as:-

Line 25 – Secondly, THE model is cross validated…
Line 26 - Our results SHOW modeling results and clinical results AS nearly matching with a smaller deviation.
Line 34 – In healthy normal’s, arteris and veins…. normal’s what?
Line 37 – HEMODYNAMIC – no “s”
Lines 37 & 38 – requires sentence restructuring
Line 38 – Figure 1 shows… – no “The”
Line 64 – major issue with the sentence
Line 68 – STRUCTURE CHANGES – NO “s” for structure
Line 75 – “…readings are seen ON THE patient…”. Sentence on this line also needs to be re-phrased.
Line 77 – Figure 4 shows… - no “The” & “…shows A PATIENT MONITOR..”
Line 80 – This procedure is commonly used to measure the pressure at various…
Line 87 – “..is A type of validation…”
Line 89 – software’s – software
Line 94 – incorrect use of the word practice
Lines 100 & 101 – sentence repetition
Line 111 – risker?
Line 112 – “…with THE help of…” & “…WE WERE able to measure…”
Line 128 – Table 1 shows modeling results AGAINST clinical measurements.
Line 130 – “modelled” is British, whereby in other places you use the American spelling. Standardize!
Line 131 – requires rephrasing
Line 137 – “…comparing Qflow results…”
Line 140 – repeats 139
Line 141 – Figure 6 shows… - no “The” & “…of A CAVM patient…” Line 142 – Table 2 compares the modeling results against Qflow results…
Line 160 – Table 3 shows….
Line 191 – The following is a list of parameters…
Line 212 – “…an accuracy of 85%.” & the following sentence needs to be re-structured
Line 213 – A snapshot…
Line 219 – Table 4 shows….
Line 220 – Location 1 and location 2 represent…
Line 243 – Collected data was submitted to the usual…
Line 244 – “..checked using Fisher’s exact…”
Line 250 – sentence very unclear
Line 268 – The author Federico…
Lines 268 & 269 – to rephrase
Line 270 – The author Kuebler…
Line 271 – The author Kienzler et al…
Line 273 - The proposed non-invasive methodology addresses the issue… what issue?
Line 275 – Our results show… - no “s” for show
Line 277 – “…patients AND 150 vessels…” – AND, not FOR
Line 279 – Statistically significant? So it is not approximately matching? Please explain.
Lines 280 to the end – requires re-work

Major, major language issues.I suggest the manuscript be sent for Professional English Editing.

---

## Round 0.4 · Major Revisions

· Academic Editor

Major Revisions

Dear Author, Please heed the suggestions given by the reviewer which mainly concern formatting. At the same time you should perform English editing of the manuscript via a professional English editor.

Reviewer 2 ·

Basic reporting

No comments

Experimental design

No comments

Validity of the findings

No comments

Additional comments

The author has not followed the format stated by peerJ for in-text citations and references. Only some part of the manuscript has been updated, but not all even though had been highlighted in previous comments.

Abstract - no results were mentioned in the results section. Author stated the methods, not displaying results. The contents are still similar as earlier submission.

Please refer to Instructions for Authors for in-text citations:-

Line 57-59 – Just mention the family name for in-text citations e.g. Kumar, Mehta & Ramachanda, 2014. I have also highlighted the wrong format of in-text citation in previous comment but still has not been corrected. References by the same author in the same year should be differentiated by letters (Smith, 2001a; Smith, 2001b).

References - please follow Instructions for Authors:-

References are still not listed according to peerJ standard even though has been advised in previous review.

For example, line 308 should be listed as –

Kumar YK, Mehta S, Ramachandra M. 2014. Cerebral Arteriovenous Malformation Modelling. Advanced Science, Engineering and Medicine. 6(1): 105-107.
The example has been given in previous comment. But the author still provide the same way which is not according to peerJ standard for line 308, line 310 and line 313.

Line 304 and 316 repeat the same reference – already mentioned in previous comment

Line 305 and 318 also repeat the same reference – already mentioned in previous comment

Line 338, 340 and 342 – list all authors here, not et.al. Each journal reference should be listed using this format: the full list of Authors with initials. Publication year. Full title of the article. Full title of the Journal, volume: page extents – already mentioned in previous comment

For example, line 338 - Federico S et.al, Mathematical models of cerebral hemodynamics for detection of vasospasm in major cerebral arteries. Acta Neurochirurgica Supplementum Volume 102, 63-69.

Missing year and not listed all authors. Should have listed:-

Cattivelli FS, Sayed AH, Hu X, Vespa P. 2008. Mathematical models of cerebral hemodynamics for detection of vasospasm in major cerebral arteries. Acta Neurochirurgica Supplementum 102: 63-69.

Please have a look for all references and follow the Instructions for Authors!

Line 69 – Yasargil et. al, 1990 is not listed in the reference list – already mentioned in previous comment.

Language issues are still the same as highlighted in previous comments – author has not done any correction as suggested.

---

## Round 0.5 · accepted · Accept

· Academic Editor

Accept

Dear Authors,

Thank you for the revised manuscript which has been accepted for publication.

Reviewer 2 ·

Basic reporting

No comments

Experimental design

No comments

Validity of the findings

No comments

Additional comments

The authors have addressed the issues raised in the earlier version of the manuscript.